# Quantification of Circulating Cell Free Mitochondrial DNA in Extracellular Vesicles with PicoGreen™ in Liquid Biopsies: Fast Assessment of Disease/Trauma Severity

**DOI:** 10.3390/cells10040819

**Published:** 2021-04-06

**Authors:** Michela Marcatti, Jamal Saada, Ikenna Okereke, Charles E. Wade, Stefan H. Bossmann, Massoud Motamedi, Bartosz Szczesny

**Affiliations:** 1Department of Neurology, University of Texas Medical Branch, Galveston, TX 77555, USA; mimarcat@utmb.edu; 2Department of Anesthesiology, University of Texas Medical Branch, Galveston, TX 77555, USA; jsaada@utmb.edu; 3Department of Surgery, University of Texas Medical Branch, Galveston, TX 77555, USA; ikokerek@utmb.edu; 4Department of Surgery, University of Texas Health Science Center in Houston, Houston, TX 77003, USA; charles.e.wade@uth.tmc.edu; 5Department of Cancer Biology, The University of Kansas Medical Center, Kansas City, KS 66160, USA; sbossmann@kumc.edu; 6Departments of Ophthalmology and Visual Sciences, University of Texas Medical Branch, Galveston, TX 77555, USA; mmotamed@utmb.edu

**Keywords:** circulating cell free DNA, traumatic brain injury, trauma severity, extracellular veciscles, PicoGreen^TM^ staining, mitochondrial DNA

## Abstract

The analysis of circulating cell free DNA (ccf-DNA) is an emerging diagnostic tool for the detection and monitoring of tissue injury, disease progression, and potential treatment effects. Currently, most of ccf-DNA in tissue and liquid biopsies is analysed with real-time quantitative PCR (qPCR) that is primer- and template-specific, labour intensive and cost-inefficient. In this report we directly compare the amounts of ccf-DNA in serum of healthy volunteers, and subjects presenting with various stages of lung adenocarcinoma, and survivors of traumatic brain injury using qPCR and quantitative PicoGreen™ fluorescence assay. A significant increase of ccf-DNA in lung adenocarcinoma and traumatic brain injury patients, in comparison to the group of healthy human subjects, was found using both analytical methods. However, the direct correlation between PicoGreen™ fluorescence and qPCR was found only when mitochondrial DNA (mtDNA)-specific primers were used. Further analysis of the location of ccf-DNA indicated that the majority of DNA is located within lumen of extracellular vesicles (EVs) and is easily detected with mtDNA-specific primers. We have concluded that due to the presence of active DNases in the blood, the analysis of DNA within EVs has the potential of providing rapid diagnostic outcomes. Moreover, we speculate that accurate and rapid quantification of ccf-DNA with PicoGreen™ fluorescent probe used as a point of care approach could facilitate immediate assessment and treatment of critically ill patients.

## 1. Introduction

The presence of extracellular DNA in plasma, serum, cerebrospinal fluid, saliva and other body fluids permits the non-invasive quantification and analysis of the DNA originating from normal and diseased cells and tissues. Blood, in particular, is easily accessible, allowing for real-time quantification and monitoring of circulating cell free-DNA (ccf-DNA) in readily available liquid biopsy samples. Although, the mechanisms by which DNA is released into the bloodstream need further investigation, currently apoptosis, necrosis, and active cellular secretion are considered as major sources of ccf-DNA, together with less known routes such as neutrophil extracellular trap release, phagocytosis, and oncosis [1,2,3]. The ccf-DNA is under extensive investigation as a biomarker for liquid biopsy aiming at early cancer detection, monitoring of disease progression, and therapeutic response [3,4,5,6,7,8,9,10]. In clinical studies, plasma DNA levels in lung cancer patients have been shown to correlate with disease stage, cancer histopathology, disease progression rate, and response to therapy [11]. Furthermore, elevated levels of circulating DNA have been also found in various non-cancer related pathologies, such as stroke, trauma, myocardial infarction, autoimmune disorders, chronic inflammation, and pregnancy-associated complications [12]. In contrast, in healthy individuals, concentration of ccf-DNA is generally found at low levels in blood [13].

Currently, quantification of ccf-DNA levels with real-time quantitative PCR (qPCR) is the gold standard for measuring DNA in blood. However, this method is dependent on primers and templates, and is also quite expensive and time consuming. In our study, we have directly compared qPCR with an inexpensive method for immediate quantification of the total amount of ccf-DNA in blood using the commercially available PicoGreen™ reagent (λ_Ex_ = 480 nm/λ_Em_ = 520 nm) (PicoGreen™ is a registered trademark of ThermoFisher Scientific, Waltham, MA, USA). Our method is advantageous in that it is primer-independent. Our goal was to show the accuracy of the PicoGreen™ method compared to the standard qPCR approach. We found strong correlation between quantification of the amount of ccf-DNA with PicoGreen™ and qPCR but only with mitochondrial DNA (mtDNA)-specific primers, but not with nuclear DNA (nuDNA)-specific primers. We also found the majority of ccf-DNA is encapsulated within extracellular vesicles (EVs) that provided protection from DNase/s that is/are present in blood. ccf-DNA was more easily detected with mtDNA-specific primers suggesting that majority of ccf-DNA is of mitochondrial origin. Finally, we concluded that quantification of ccf-DNA with PicoGreen™ could provide immediate critical clinically relevant information that may be particularly useful in emergency room settings.

## 2. Materials and Methods

### 2.1. Human Subjects

In this study we used serum obtained from the Lung Cancer Biospecimen Resource Network (LCBRN). The LCBRN is a network of three academic medical centers: the Medical University of South Carolina (MUSC, Columbia, SC, USA), the University of Virginia (UVA, Charlottesville, VA, USA), and Washington University in St. Louis (WUSTL, Dr, St. Louis, MO, USA). Biospecimens were collected at these sites according to standard operating procedures and were shipped to the LCBRN coordination center at UVA for storage. LCBRN is an open access biorepository that provides specimens to academic and private industry scientists worldwide. The experimental protocol was approved by the UTMB Institutional Review Board (IRB), and this study was conducted in compliance with ethical and safe research practices involving human tissues. In this study we also used serum of adult patients that were admitted to a level 1 trauma center from July 2011 to May 2016 at the University of Texas Health Science Center at Houston (UTHSC, Houston, TX, USA). The study was approved by the UTHSC at Houston IRB. Adult patients who were admitted to our hospital and who required trauma team activation were eligible for inclusion. Patients were excluded if they were pregnant, were prisoners, were enrolled in other studies, declined to consent, or if no blood sample was drawn on admission. Consent was obtained from the patient or a legally authorized representative within 72 h of admission or waived for patients who were discharged or died within 24 h of hospital admission. No changes in clinical practice were implemented in this observational study. Samples were also collected at the University of Texas Medical Branch under protocol approved by the IRB. The inclusion criteria of subjects include confirmed diagnosis of primary lung cancer, chronic bronchitis or asthma, and written informed consent from subject exclusion. Exclusion criteria includes pregnant status and prisoners.

### 2.2. DNA Isolation and qPCR

Total DNA from 100 μL serum was isolated using the DNeasy Blood & Tissue Kit from Qiagen (Germantown, MD, USA) with a final elution volume of 100 μL. The isolated DNA was amplified by using the Maxima SYBR Green/ROX qPCR Master Mix (Thermo Scientific, Waltham, MA, USA) in final volume of 10 μL (4.4 μL of DNA, 0.6 μL 10 μM primers, 5 μL of master mix) with the following primers:mtNAD1: FW 5′-ATACCCATGGCCAACCTCCT-3′, RV 5′-GGGCCTTTGCGTAGTTGTAT-3′;mtCOXIII: FW 5′-TGACCCACCAATCACATGC-3′, RV 5′-ATCACATGGCTAGGCCGGAG-3′;mtCYTB: FW 5′- ATGACCCCAATACGCAAAAT-3′, RV 5′- CGAAGTTTCATCATGCGGAG-3′;nuSIRT1: FW 5′-CCCGCAGCCGAGCCGCGGGG-3, RV 5-TCTTCCAACTGCCTCTCTGGCCCTCCG-3′;nuACTB: FW 5′-CATGTACGTTGCTATCCAGGC-3′, RV 5′-CTCCTTAATGTCACGCACGAT-3′

We used the following thermal cycle: 95 °C for 10 min, 40 cycles at 95 °C for 15 s, and 60 °C for 1 min. Each reaction was run in duplications. We calculated amount of mtDNA using three different mtDNA-specific genes (CYTB, NAD1, and COXIII) and amount of nuDNA using two different nuclear genes (SIRT1, and ACTB). The amount of ccf-DNA was calculated based on Ct values and previously generated standard curve. For this we used different know concentration of isolated mtDNA followed by qPCR with mtDNA-specific primers. We also used different concentration of total DNA followed by qPCR with nuDNA-specific primers (we did not used isolated nuDNA since total DNA contain less than 5% of mtDNA). Both approaches generated similar data where for each DNA concentration (in the range from 1 pg to 10 ng) specific Ct value was obtained.

### 2.3. DNA Quantification Using PicoGreen Reagent

The amount of total DNA in cell free serum was measured using Quanti-iT^TM^ PicoGreen™ (ThermoFisher Scientific). Briefly, 20 μL of cell free serum was mixed with 30 μL of PBS, followed by addition of 50 μL of PicoGreen™ reagent (5 μL of Quanti-iT^TM^ PicoGreen™ in 1 mL of PBS). The fluorescence was measured (Ex 480/Em 520) on a SpectraMax M2e spectrophotometer (Molecular Devices, San Jose, CA, USA).

### 2.4. DNase Treatment and Isolation of Extracellular Vesicles (EVs)

For DNase treatment, 20 μL of cell free serum was incubated with 10 mU of recombinant DNase (Qiagen) for 30 min at 37 °C followed by detection with PicoGreen™ reagent. EVs from serum were isolated using the Total Exosome Isolation Kit from serum (Invitrogen, Carlsbad, CA, USA) according to the manufacturer’s recommendations.

### 2.5. Statistical Analysis

All statistical analysis was performed using GraphPad Prism software (GraphPad Software, San Diego, CA, USA). Data distribution was first assessed using normality testing. Since each analyzed dataset had at least one groups which was not normally distributed, we used Kruskal-Wallis tests to determine significance. Significance differences are denoted as: * *p* < 0.05, ** *p* < 0.01, *** *p* < 0.001, **** *p* < 0.0001.

## 3. Results

### 3.1. Increased Amounts of ccf-DNA in Serum of Human Lung Adenocarcinoma Subjects

We analyzed cell free serum samples of a group of patients diagnosed with various stages of lung adenocarcinoma (*n* = 57) and a group with benign growth (*n* = 20) that we received from the Lung Cancer Biospecimen Resource Network (https://lungbio.sites.virginia.edu). The basic demographic information of both groups is provided in Table 1. Total DNA in cell free serum was isolated followed by qPCR with several sets of primers specific for: the mitochondrial cytochrome c oxidase subunit III (mtCOXIII) gene, the mitochondrial NADH dehydrogenase subunit I (mtNADI) gene, the mitochondrial cytochrome b (mtCYTB) gene, the nuclear actin beta (nuACTB) gene and the nuclear sirtuin 1 (nuSIRT1) gene. The amount of DNA was calculated based on obtained ΔCt values that were compared to standard curves generated using known concentration of DNA and expressed as pg of DNA in 100 μL of serum. Obtained results were compared with the amount of DNA present in serum samples of healthy volunteers (*n* = 20) using the same set of primers.

We did not find any differences in the amount of mtDNA when comparing the control group with subjects with benign tumors, using three sets of mtDNA-specific primers (Figure 1A and Appendix AA). However, when comparing the control group to subjects with benign tumors, there was an increase of nuDNA when using nuACTB primers but a decrease when using nuSIRT1 primers (Figure 1B and Appendix AB). Next, we compared the amount of ccf-DNA among the control group, subjects with benign tumors and subjects with adenocarcinoma of the lung. Compared to the control group and the subjects with benign tumors, a significant increase of both mtDNA (mtCOXIII) and nuDNA (nuACTB, nuSIRT1) was seen in subjects with adenocarcinoma (Figure 1A,B and Appendix AB). However, the three groups had no differences in calculated ccf-DNA when two other mtDNA-specific primers, mtNADI and mtCYTB, were used (Appendix AA). We also found a significant difference between the amount of the calculated DNA with mtDNA- and nuDNA-specific primers. The amount of mtDNA was two to three orders of magnitude higher in comparison to nuDNA (Figure 1 and Appendix A). Potentially, the high amount of mtDNA in comparison with nuDNA could be explained by the presence of the multiple mitochondria, with multiple copies of mtDNA in each cell in comparison of two copies for each nuclear gene. An alternative explanation is that there was selective mtDNA release to bloodstream.

Next, we compared mtDNA vs. nuDNA contents in the serum of subjects with various stages of the adenocarcinoma of the lung. Regardless of primer location, there was no difference in ccf-mtDNA or ccf-nuDNA levels at different stages of adenocarcinoma of the lung (Figure 1C,D and Appendix AC,D). Our data also revealed minimal variation in the amount of calculated mtDNA in serum in all analyzed sample groups regardless of primers location. But we measured differences of up to three orders of magnitude in the amount of nuDNA calculated with different nuDNA-specific primers (Table 2). The large difference between amounts of mtDNA and nuDNA, as well as the large variation among nuDNA-specific primers, suggests either unequal or specific representation of nuDNA in the serum. This result also confirms the qPCR as the primer and template-specific method, and that ccf-DNA could be easily detected with mtDNA-specific primers.

It should be noted that the data obtained up to this point were generated using serum samples that were stored (but never thawed and frozen) for an extended period of time (years), which may affect the stability of DNA. In order to test for potential effects on ccf-DNA stability by storing samples at −80 °C, we obtained fresh blood of healthy volunteers (*n* = 4) and subjects diagnosed with adenocarcinoma of the lung (*n* = 12) from the Division of Cardiothoracic Surgery at UTMB. Although we did not perform statistical analysis of these samples due to the low number of control subjects (*n* = 4), our results were similar to the results obtained with the LCBRN samples which were stored at −80 °C for extended period of time (Figure 1 and Appendix A). We also saw in the fresh samples that there was an increase amount of ccf-DNA in the serum of adenocarcinoma subjects when compared to the control group using mtDNA-specific primers (Figure 2A) and nuDNA-specific primers (Figure 2B). Also, as before, we found a marked difference in the amount of nuDNA when analyzed with nuACTB and nuSIRT1 primers (Figure 2B). Thus, we showed that similar results could be obtained with fresh blood compared to blood serum stored at −80 °C. Overall our data showed that there is a significant increase of ccf-DNA in the blood of adenocarcinoma subjects compared to control groups. But results obtained by qPCR can vary significantly depending on the primer and template used.

### 3.2. Detection of Circulating Cell-Free DNA with PicoGreen Reagent

Our studies clearly showed a significant increase of both mtDNA and nuDNA in subjects with adenocarcinoma of the lung when compared to healthy controls. Our data also indicated that analysis of blood with qPCR can be primer and template specific, particularly for nuDNA. Based on these results, we decided to test a fast and cost-effective, primer and template independent method to detect ccf-DNA in blood samples. We used PicoGreen™ reagent (λ_Ex_ = 480 nm/λ_Em_ = 520 nm), which has been shown to detect as little as 25 pg/ mLof dsDNA in the presence of RNA, and free nucleotides. The amount of ccf-DNA was calculated using 20 μL of cell free serum obtained from blood of healthy volunteers (*n* = 4) and subjects diagnosed with lung adenocarcinoma (*n* = 12) from the Division of Cardiothoracic Surgery at UTMB using a standard curve that was generated using Λ DNA. In agreement with qPCR data (Figure 2), we measured significant increases of DNA in the serum of lung adenocarcinoma subjects (Figure 3A). It should be noted that the PicoGreen™ assay works instantaneously and that the fluorescence intensity remains constant after mixing. In addition, the amount of quantified ccf-DNA in both control groups and lung adenocarcinoma subjects using PicoGreen™ reagent closely resembled the amount of ccf-DNA measured with mtDNA-specific primers (Figure 2A and Figure 3A). These results indicate that quantification of ccf-DNA using mtDNA-specific primers closely resembles the amount of total ccf-DNA and suggests that the majority of ccf-DNA is of mitochondrial origin.
Figure 2Subject with lung adenocarcinoma have high amount of ccf-DNA. The amount of mtDNA (**A**) and nuDNA (**B**) was quantified with qPCR using mtCOXIII and nuACTB and nuSIRT1 primers, respectively, in serum of healthy volunteers (*n* = 4) and subjects with lung adenocarcinoma (*n* = 12). The serum was obtained at UTMB clinics. The amount of DNA is expressed as pg of DNA in 100 μL of serum.
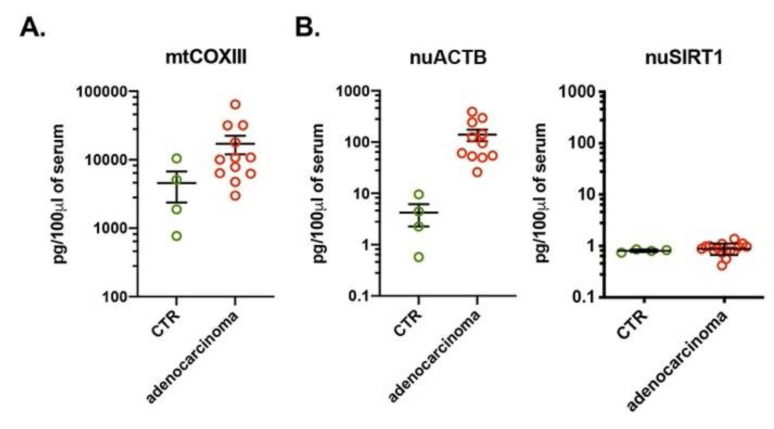



### 3.3. Ccf-DNA in Blood Is Present within Extracellular Vesicles (EVs)

One major concern when measuring DNA concentrations in whole blood is the presence of DNase I, which may affect the stability of ccf-DNA. DNase I is a pancreatic enzyme that is present in circulation and capable of degrading free floating DNA [14]. To confirm the presence of active DNase I in serum we incubated 30 ng of Λ DNA with 40 and 80 μL of fresh serum for 30 min at 37 °C and measured 14% and 27% decrease of the DNA signal measured with PicoGreen™ reagent, respectively (Figure 3B). These data confirmed the expected presence of active DNase I in the serum. Next, we incubated the serum of lung adenocarcinoma subjects with 10 mU of recombinant DNase for 30 min at 37 °C followed by detection with PicoGreen™ reagent. The analysis of the serum of three lung adenocarcinoma subjects showed that an average 90% of the ccf-DNA present in serum is DNase insensitive (Figure 3C). Since 90% of the DNA present in the serum is protected from DNases, we hypothesize that ccf-DNA is localized within lumen of EVs. It is known that serum, plasma, and other liquid biopsies contain membranous EVs derived from various cell types. EVs participate in physiological and pathological processes and have potential applications in diagnostics and therapeutics. EVs are typically classified into exosomes, microvesicles and apoptotic bodies [15,16]. Since DNA in serum is DNase(s) insensitive, we hypothesize that ccf-DNA is encapsulated within exosomes and/or microvesicles. To test this hypothesis, we isolated EVs from 25 μL of plasma from two lung adenocarcinoma subjects and one healthy control using a Total Exosome Isolation kit. Note, that isolation of EVs using the precipitation reagent presented in the kit does not separate exosomes from microvesicles [17]. Next, the amount of DNA was analyzed by qPCR using mtCOXIII and nuACTB primers. We compared the amount of DNA in input (serum), isolated EVs and in supernatant that represented free-floating DNA after the EVs were isolated. Similar to our previous data, we detected the presence of mostly mtDNA with negligible amounts of nuDNA in both serum and isolated EVs (Figure 3D,E). As expected, the amount of DNA was higher in adenocarcinoma samples compared to the control group (Figure 3D,E). Moreover, more than 90% of ccf-DNA in serum was present within EVs, with negligible amount of free-floating DNA (Figure 3D,E). These data further support our assumption that free-floating DNA are degraded by the active DNase I present in blood but protected by the bilayer membrane of EVs. Most importantly, the majority of ccf-DNA within EVs can be efficiently detected using mtDNA-specific primers and thus may be of mitochondrial origin. Most importantly, ccf-DNA within EVs can be quantified with the PicoGreen™ reagent as a fast and inexpensive alternative method to qPCR.
Figure 3Circulating extracellular vesicles (EVs) contain a majority of the ccf-DNA that has mitochondrial origin. (**A**) PicoGreen quantification of ccf-DNA in serum of healthy volunteers (*n* = 4) and subjects with lung adenocarcinoma (*n* = 12) admitted to UTMB’s clinic. (**B**) Serum contain active DNase/s. Changes in the fluorescent units of Λ DNA incubated with increasing volume of serum (40 and 80 μL). (**C**) Ccf-DNA is DNase insensitive. Serum samples of three lung adenocarcinoma subjects were incubated with recombinant DNase and changes in DNA concentration measured with PicoGreen™ expressed in fluorescent units are shown. Ccf-DNA is localized within lumen of EVs. Comparison in the amount of DNA between input, serum after EVs isolation (supernatant) and isolated EVs (pellet) of three subjects using (**D**) mtCOXIII and (**E**) nuACTB primers.
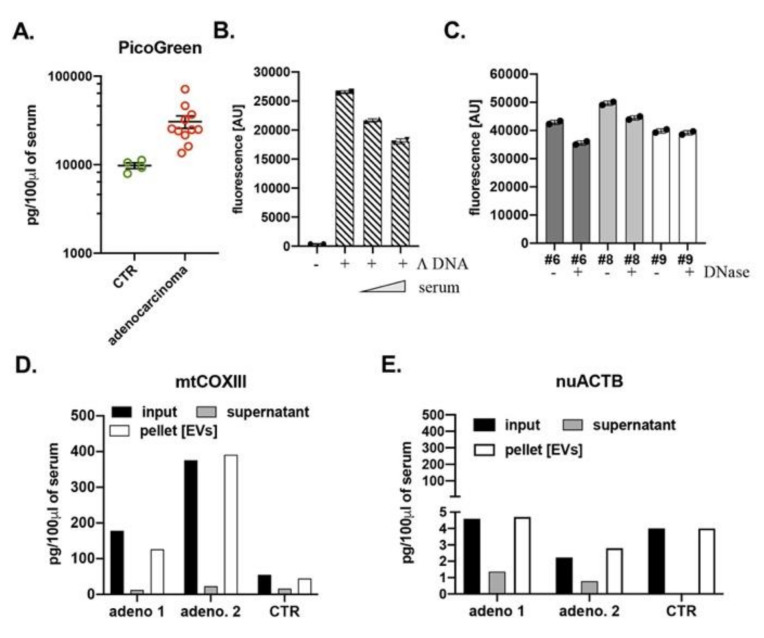


### 3.4. Increased Amount of ccf-DNA in Plasma of Traumatic Brain Injury Subjects

To confirm that PicoGreen measurement of ccf-DNA can be an alternative method to the gold standard qPCR for measuring ccf-DNA, we tested serum samples of healthy volunteers (*n* = 20) and subjects with non-penetrating traumatic brain injury (TBI; *n* = 53) that were admitted to the hospital of the UTHSC at Houston. The TBI subjects consisted of 20 patients with mild TBI (mTBI) and 33 patients with severe TBI (sTBI) as assessed by the Glasgow Come Scale (GCS): GCS 13-15 for mTBI, GCS < 8 for sTBI [18]. Importantly, the serum of TBI subjects were harvested immediately after admission to the intensive care unit when GCS was not known. The total amount of DNA was analyzed using qPCR with mtDNA and nuDNA specific primers as was done with the adenocarcinoma subjects. The amount of mtDNA calculated with the mtCOXIII and mtNADI primers was significantly increased in both TBI groups compared to the control group, and the amount of calculated mtDNA was very similar using both primers (Figure 4A). There was no significant difference when comparing mTBI patients to sTBI patients. However, when using either nuDNA-specific primer (nuACTB and nuSIRT1), we measured a significant increase of nuDNA when comparing the control group to sTBI patients (Figure 4B).

Similar to our experiments in adenocarcinoma subjects, there was a significant difference (at least an order of magnitude) in the amount of calculated nuDNA between nuACTB and nuSIRT1 primers (Figure 3B). Interestingly, the amount of mtDNA calculated using two different primers (mtCOXIII and mtNADI) were nearly identical. In addition, the amount of calculated mtDNA was significantly higher when compared to the amount of nuDNA.

Finally, we measured the amount of ccf-DNA in control and TBI subjects using PicoGreen™ reagent. As with qPCR, the amount of ccf-DNA was significantly increased in both TBI groups compared to the control group. The amount of calculated ccf-DNA with PicoGreen™ was very similar to ccf-DNA measured with qPCR using mtDNA-specific primers (Figure 4C). In addition, using the PicoGreen™ reagent we were able to detect significant differences between mTBI and sTBI (Figure 4C). To verify direct correlation between amount of ccf-DNA in TBI subjects calculated with PicoGreen™ and qPCR using mtDNA-specific and nuDNA-specific primers, we performed linear regression analysis. As shown in Appendix A we detected significant correlation between ccf-DNA quantified with PicoGreen™ and mtCOXIII (R^2^ = 0.41) but not between PicoGreen™ and nuACTB (R^2^ = 0.02). Together, these data further confirm feasibility of quantification of ccf-DNA using PicoGreen™ reagent as a fast and cost-effective alternative method to qPCR.

## 4. Discussion

Significant changes in the amount of DNA in liquid biopsies have been demonstrated in several types of cancer [19,20,21,22,23], but also various forms of trauma, such as TBI [24], stroke [25], cardiovascular diseases [26] or acute respiratory distress syndrome [27]. Changes in the concentration of circulating DNA have been shown to have therapeutic and prognostic values. However, the majority of reported analyses utilized qPCR, a technique that is primer- and template-dependent, time consuming and not cost-effective method. In this report, we proposed and validated a fast and cost-effective method for quantifying the total amount of ccf-DNA in blood samples using the fluorescent PicoGreen™ reagent.

Quantification of DNA in body fluids using fluorescent probes has been proposed previously. The SYBR^®^Gold stain was shown to be useful for quantification of DNA contents in several bodily fluids [28]. Although this study provided technical evidence of the possibility of measuring ccf-DNA in bodily fluids with fluorescent probes, it was not performed in the context of disease diagnosis and staging. More recently, the amount of ccf-DNA in subjects with breast cancer was also analyzed with the SYBR^®^Gold fluorescent probe [29]. It showed a good correlation between the amount of ccf-DNA and the diagnosed stage of breast cancer, but no direct comparisons with qPCR was performed. In this study, we evaluated the potential use PicoGreen™ as fluorescent dye for the quantitative measurement of ccf-DNA in liquid biopsies. PicoGreen™ reagent (λ_Ex_ = 480 nm/λ_Em_ = 520 nm) can detect as little as 25 pg/ mL of double stranded DNA (dsDNA) in the presence of RNA, and free nucleotides. Whereas PicoGreen™ does not exhibit a significant background fluorescence signal, its fluorescence is switched on upon dsDNA binding [30]. A recent study has demonstrated that PicoGreen™ acts as both, minor-groove binder and DNA intercalator [31]. The assay is linear over three orders of magnitude (1 to 1000 ng dsDNA mL^−1^) and has no sequence dependence, thus allowing accurate measurement of DNA in various liquid biopsies and tissue extracts. PicoGreen™ staining has been extensively used for histochemistry staining of both nuDNA and mtDNA. It has been used in imaging changes, e.g., condensed nuDNA structures with super-resolution fluorescent microscopy in live time [32], or for detection of mtDNA depletion in cultured cells [33]. Recently, PicoGreen™ staining was utilized in the quantitative investigation of DNA in plasma of mice subjected to total body irradiation. The amount of ccf-DNA correlated with the total radiation dose [34]. Most importantly, in opposite to SYBR^®^Gold stain, PicoGreen™ does not detect single-stranded DNA or RNA but only dsDNA that is the best template for qPCR and thus can be directly compared to qPCR.

From a clinical standpoint, the ability to use PicoGreen™ to measure ccf-DNA in lung cancer patients would address a major gap in care of these patients currently. Unfortunately, lung cancer is highly morbid and is generally diagnosed at a late stage. Nearly 80 percent of patients with lung cancer are diagnosed as Stage III or IV, and less than 20 percent live more than 5 years after diagnosis [35]. Screening with computed tomography (CT) scan in appropriate patients can reduce mortality from lung cancer. But less than 5 percent of patients who should be screened ever receive a screening CT scan [36]. Utilizing a more cost-effective screening modality would be able to reach more patients and potentially have a greater effect in reducing mortality from lung cancer. The above experiments show that measurement of ccf-DNA levels with the PicoGreen™ method may be a clinical tool which could be applied during routine physical examination and the PicoGreen™ method is advantageous compared to the qPCR method in that it is much less expensive and could be used in widespread fashion without increasing the overall cost of health care significantly.

Although our studies showed no correlation with stage of lung cancer, our study had a relatively limited number of patients. Since the PicoGreen™ method is so inexpensive, future studies should be performed to examine the correlation of ccf-DNA levels and various characteristics of patients with lung cancer. Clinical and pathologic variables such as tumor diameter, number of lymph nodes involved and presence of lymphovascular invasion may be shown to correlate with ccf-DNA levels if a large group of lung cancer patients are analyzed. The ramifications of such correlations would be the ability to guide treatment decisions based on ccf-DNA levels using the inexpensive PicoGreen™ method. Unfortunately, many patients with Stage I lung cancer recur and ultimately die of disease. If there were a better ability to stratify these patients with an inexpensive test, then high-risk Stage I patients could receive closer surveillance and earlier chemoradiation treatment. There is currently no accepted serum test which has identified a high-risk subset of Stage I lung cancer patients, but use of the PicoGreen™ method may ultimately solve another gap in clinical understanding that is currently present.

Our data show a close correlation with the amount of ccf-DNA measured by PicoGreen™ and qPCR with mtDNA-specific primers. These suggest that the majority of ccf-DNA have mitochondrial origin. We obtained similar amounts of mtDNA in serum calculated using three different sets of primers that are located in three distal regions of mtDNA. This is in opposite to quantification of nuDNA using two different set of primers where three orders of magnitude differences were measured between nuACTB and nuSIRT1-specific primers, even there are only two copies of each nuclear gene per cell. This suggests unequal distribution of ccf-nuDNA and/or specific release to blood stream of mtDNA. The latter is further supported by our data showing the presence of mostly mtDNA in EVs in serum of analyzed samples. It has been previously shown that more than 90% of ccf-DNA is in fact associated with small EVs: exosomes and microvesicles [37]. At the present, very little is known of how DNA is packaged into EVs. The best known mechanism for DNA to be loaded into EVs involved their biogenesis that included encapsulation of the cytosolic DNA within lumen of EVs [16]. The presence of particular mtDNA in cytoplasm has been well documented in various pathological conditions, including infection [38,39], neurodegeneration [40], and cancer [41], to name just few. The presence of mtDNA in EVs has been reported [42,43]. Apoptotic cell death can result in generation of circulating apoptotic bodes, the largest known EVs, possessing nuclear DNA. However, the mechanism of translocation of nuclear DNA or fragments of nuclear DNA to the cytoplasm that can be loaded into small EVs (exosomes and microvesicles) are largely unknown. One of the proposed mechanisms involves formation of micronuclei, a nuclear-enclosed structure that originated from chromatid fragments caused by misrepaired/unrepaired DNA breaks or malsegregations of chromosomes [44,45,46]. Our data strongly indicate that majority of ccf-DNA has mitochondrial origin that is present in small EVs but still we cannot exclude the possibility that these reflect the presence of several more copies of mtDNA then nuDNA in each cell. Also, both of our methods for quantification of ccf-DNA detect dsDNA and primers that were used for qPCR detects DNA fragments of a size of at least 100–200bp. Thus, we cannot exclude possibility that the amounts of particularly ccf-nuDNA as a free-floating and/or within EVs is underestimated since due to the large size of the nuclear DNA it may be more sensitive to digestion by DNases that can generate short ssDNA fragments that are not detected by PicoGreen™ and qPCR. Thus, further investigation are needed such as DNAseq or alternatively by preforming DNA repair assay prior qPCR/PicoGreen™. Nerveless, we showed that ccf-DNA could be much easily detected using qPCR with mtDNA-specific primers that closely correlated with measured of ccf-DNA with PicoGreen™ reagent.

One concern that has not been previously taken under consideration is the presence of active DNases in the circulation. The presence of pancreatic Dnase I has been known for a long time [14]. Our data support the notion about the presence of active DNase(s) in human serum (Figure 3). It can be assumed that active Dnase I will degrade free floating DNA and thus, the quantification of ccf-DNA may change over the time, which can greatly diminish the diagnostic value of the data. Our data provide evidence that in freshly analyzed serum samples the major fraction of the DNA is present within the lumen of EVs and thus is DNase insensitive. Although, our observation requires additional analysis of mtDNA-carrying EVs, such as EVs’ size and the presence of the specific markers, it is in agreement with recent reports showing that most ccf-DNA in human plasma is localized within EVs [37]. Since enhanced oxidative stress is directly linked with most of the pathologies including cancer and various forms trauma, we hypothesize that mtDNA is specifically released from cells upon injury. We have recently shown that oxidative stress causes selective release of oxidatively damaged mtDNA to extracellular space via EVs in cultured lung epithelial cells [42]. The diagnostic utilization of EVs in several pathologies “exploded” in recent years [47,48,49]. However, mtDNA in EVs as a biomarker has not been carefully evaluated. Moreover, the presence of ccf-DNA within EVs opens several opportunities for future studies. An advantage of ubiquitous presence of EVs in liquid biopsies is that they can cross vascular barriers, such as the blood-brain barrier, and can be detected in circulation [50,51]. Thus our future studies will include investigation of the brain-specific markers in circulating EVs post TBI. The increasing number of reports investigate cell/tissue type specific markers present in EVs that in the future could lead to identification of circulating EVs’ source/s.

Finally, our data with TBI subjects showed that quantification of ccf-DNA could provide clinically relevant information about severity of trauma. We speculate that this approach could be particularly useful in emergency settings where resources are scatter but immediate information about trauma severity could provide important information from a triage and resource standpoint.

## Figures and Tables

**Figure 1 cells-10-00819-f001:**
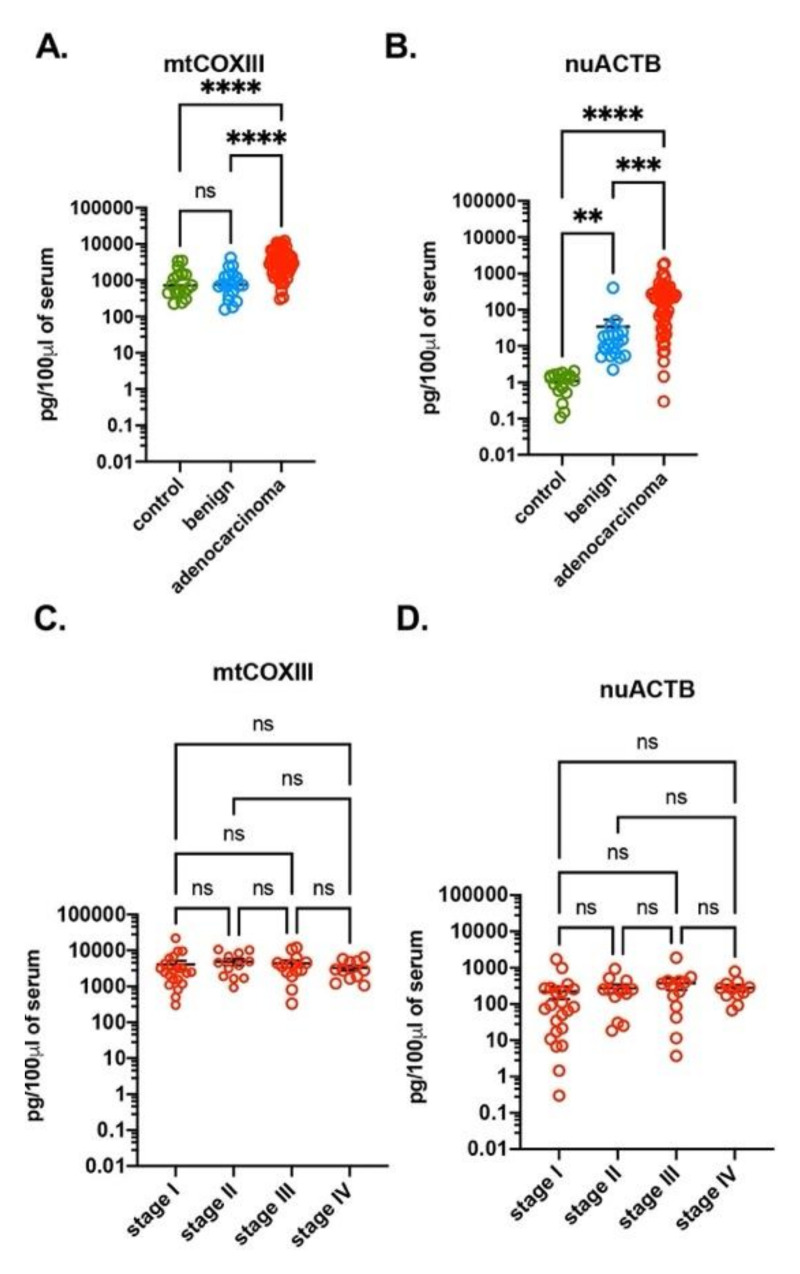
Serum of lung adenocarcinoma subjects have a high amount of ccf-DNA. The amount of mtDNA (**A**) and nuDNA (**B**) were quantified with qPCR using mtCOXIII and nuACTB primers, respectively, in serum of healthy volunteers (*n* = 20), subjects with benign tumors (*n* = 20) and with lung adenocarcinoma (*n* = 57). The amount of mtDNA (**C**) and nuDNA (**D**) were quantified with qPCR using mtCOXIII and nuACTB primers, respectively, in serum of subject with stage I (*n* = 21), stage II (*n* = 12), stage III (*n* = 13) and stage IV (*n* = 11) of lung adenocarcinoma. The serum was obtained from LCBRN. The amount of DNA is expressed as pg of DNA in 100 μL of serum. ** *p* < 0.01, *** *p* < 0.001, **** *p* < 0.0001.

**Figure 4 cells-10-00819-f004:**
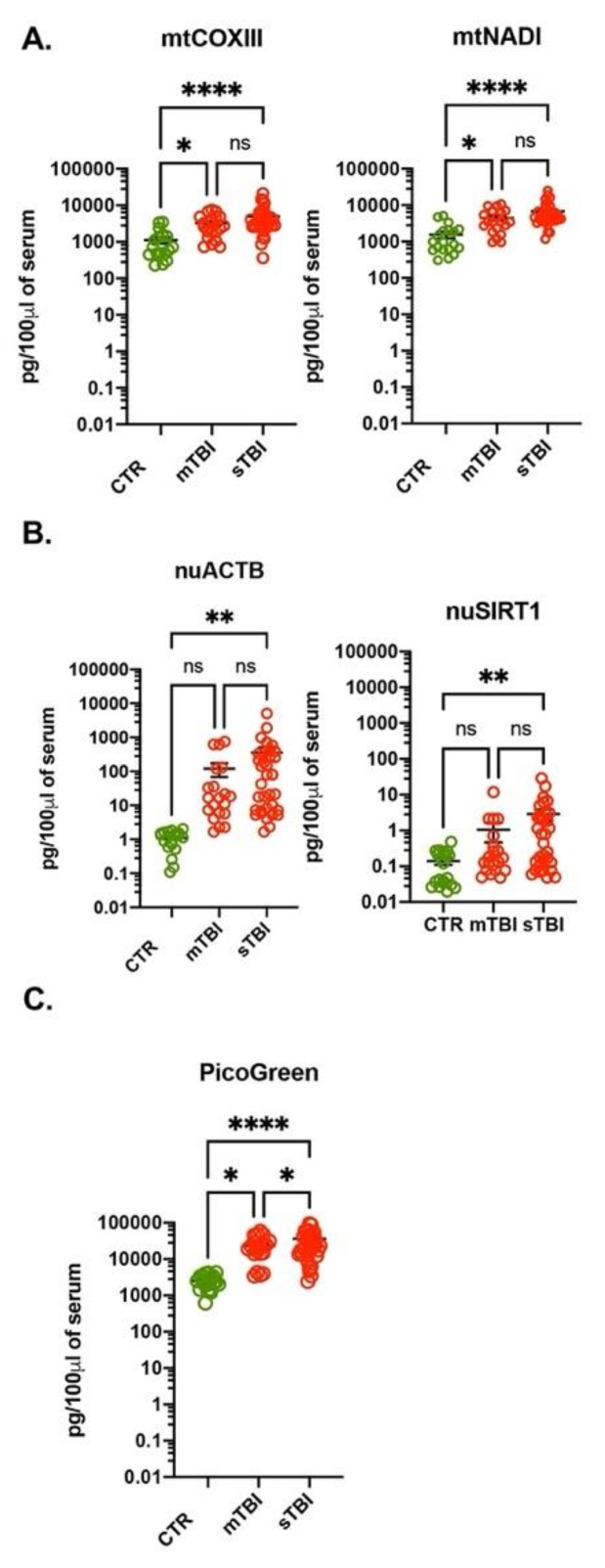
Traumatic brain injury patients have high amount of ccf-DNA. The amount of mtDNA (**A**) and nuDNA (**B**) was analyzed with qPCR using mtCOXIII, mtNADI and nuACTB and nuSIRT1 primers, respectively, in serum of healthy volunteers (*n* = 20), mTBI (*n* = 20) and sTBI (*n* = 33). (**C**) The amount of ccf-DNA measured with PicoGreen in healthy volunteers, mTBI and sTBI. The amount of DNA is expressed as pg of DNA in 100 μL of serum. * *p* < 0.05, ** *p* < 0.01, **** *p* < 0.0001.

**Table 1 cells-10-00819-t001:** Demograpy of patients with benign and with lung adenocarcinoma tumors.

		Benign *n* = 20	Lung Adenocarcinoma *n* = 57
gender	male	11	26
	female	9	31
age (years)	mean ± SD	59 ± 10	67 ± 10
	range	42–73	33–92
smoking history	never	6	7
	quit	7	35
	current	7	15
tumor stage	I		21
	II		12
	III		13
	IV		11

**Table 2 cells-10-00819-t002:** DNA in serum of subjects with lung adenocarcinoma (pg/100μL).

	mtCOXIII	mtCYTB	nuGAPDH	nuSIRT1
average	4187.52	3567.89	269.94	0.34
SEM	501.34	1104.41	47.87	0.03

## Data Availability

The data presented in this study are available on request from the corresponding authors.

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
