# Peer review of "Quantification of Circulating Cell Free Mitochondrial DNA in Extracellular Vesicles with PicoGreen™ in Liquid Biopsies: Fast Assessment of Disease/Trauma Severity"

_cells, 2021, doi:10.3390/cells10040819_

Round 1

Reviewer 1 Report

1. This manuscript still need a complete proofreading in English grammar and some spelling. For example, misreparied should be misrepaired in Lane 350.
2. The major defect of this research is that the PicoGreen staining is a commmon method in research of cfDNA/ccf-DNA or other cell free DNA fragments. The result is not novel as a method development article. I encourage the author modify the paper to be a clinical validation article for publication.
3. The qPCR and hybridization only work well on intact DNA/RNA templates. However, the cell free DNAs are always partially degraded or damaged by enzymes within cells or in serum. This had been verified by DNA repairing and DNAseq fragmentation analysis. The author only used qPCR to calculate some intact fragment so drawed an improper conclusion.
4. The author must perform DNA repair before qPCR or DNAseq library preparing. The circular mtDNAs are relatively abundant and more intact. While the small gDNA (less than 100bp or smaller) only can be calculated by DNAseq. So the author can emphasize the existance of ccf-DNA in EVs but can not ignore the existance of ccf-DNA outside of EVs, based on their current methods and results.
5. 2.2 DNA isolation and RT-qPCR is not correct. The author only perform qPCR on DNA, not RT-qPCR on RNA.
6. The Discussion is redundant and should be simplified. The author may discuss more on clinical validation, not the method development.

Author Response

  1. This manuscript still need a complete proofreading in English grammar and some spelling. For example, misreparied should be misrepaired in Lane 350.

Response: We apologize for any inconvenience. The revised manuscript was carefully edited, including complete proofreading for English grammar and spelling.

  1. The major defect of this research is that the PicoGreen staining is a common method in research of cfDNA/ccf-DNA or other cell free DNA fragments. The result is not novel as a method development article. I encourage the author modify the paper to be a clinical validation article for publication.

Response: We appreciate the observation made by the reviewer recognizing the potential for clinical applications of the methodology described in our manuscript. Furthermore, although we agree with the reviewer that the use of PicoGreen as a reagent for measuring DNA concentration has been known for a long time, we are excited about the application of this approach as a novel tool for quantification of any kind extracellular double-stranded ccf-DNA, leading to profiling and quantification of biomarkers of tissue injury and disease progression that may be present in liquid biopsies. To further address this concern of the reviewer, we conducted additional PubMed searches revealing that with the exception of one small pilot clinical study involving 10 patients suffering from lung cancer, all the other studies using our proposed approach were conducted in cells and under in vitro conditions. In agreement with the reviewer’s recommendation, we believe that our report could serve as the first comprehensive study using clinical samples analyzing extracellular DNA demonstrating the advantages of using a fluorescent probe (PicoGreen) as a rapid and inexpensive assay for accurate quantification of extracellular DNA compared to using golden standard method of qPCR in two distinct sets of pathological conditions (lung cancer and traumatic brain injury). Therefore, in agreement with the reviewer and in response to his constructive comments, we modified our manuscript emphasizing the potential for the clinical application of our approach detailing the studies that we have done validating the diagnostic power of our approach using clinical samples.

  1. The qPCR and hybridization only work well on intact DNA/RNA templates. However, the cell free DNAs are always partially degraded or damaged by enzymes within cells or in serum. This had been verified by DNA repairing and DNAseq fragmentation analysis. The author only used qPCR to calculate some intact fragment so drawed an improper conclusion. The author must perform DNA repair before qPCR or DNAseq library preparing. The circular mtDNAs are relatively abundant and more intact. While the small gDNA (less than 100bp or smaller) only can be calculated by DNAseq. So the author can emphasize the existance of ccf-DNA in EVs but can not ignore the existance of ccf-DNA outside of EVs, based on their current methods and results.

Response: We are thankful for this comment. We agree with the reviewer that some portion of the DNA is present outside of the EVs and may not be detected by both qPCR and PicoGreen due to its small size and quality. Moreover, due to the presence of active DNases in the blood, the amount and size of “free-floating” DNA will change over time and thus may not be particularly useful as a clinical tool. Our data show that most of the DNA that could be detected by qPCR and PicoGreen is within lumen of extracellular vesicles and is therefore protected from circulating DNases. We agree that DNA repair assay or DNAseq should be performed to investigate the amount and size of free-floating DNA, but this is outside the major scope of the current manuscript. Nevertheless, we have recently showed that low levels of oxidative stress (that is believed as an underlaying cause of most of the pathologies including lung cancer and traumatic brain injury) causes the release of fragments of mtDNA via EVs. We have estimated that ~3kb fragments of the mtDNA together with negligible amounts of nuDNA are released via EVs from oxidatively stress lung epithelial cells (DOI: 10.1038/s41598-018-19216-1). The major goal of the current report is to directly compare qPCR (with nu and mtDNA-specific primers) with PicoGreen resulting a conclusion that fast and accurate measurement of ccf-DNA using PicoGreen staining provides comparable qPCR results. This is further supported with additional analysis that we included in the revised manuscript showing direct correlation between PicoGreen and qPCR with mtDNA-specific primers (Figure S2). We concluded that measuring ccf-DNA using PicoGreen could be particularly useful in the emergency room setting where fast assessment of ccf-DNA may provide clinically important information about trauma severity (e.g. TBI) in the challenging environment. Based on the reviewer comments, in the revised manuscript, we have addressed the limitation of our approach in particular discussing the possibility of under detection of free-floating DNA, DNA quality and size in our analytical approach.

  1. 2.2 DNA isolation and RT-qPCR is not correct. The author only perform qPCR on DNA, not RT-qPCR on RNA.

Response: We apologize for using incorrect abbreviation. The mistake came from the first version of the manuscript where the RT-qPCR abbreviation was used for a real-time quantitative PCR that we subsequently corrected for qPCR as a more appropriate since RT-qPCR is more appropriate abbreviation for revers-transcription quantitative PCR. In the revised manuscript we have corrected this mistake.

  1. The Discussion is redundant and should be simplified. The author may discuss more on clinical validation, not the method development.

Response: We are thankful for this suggestion. Per reviewer suggestion, the revised manuscript is keenly focus on describing and discussing the clinical validation of our work. We have also expanded the clinical implications of our findings. We feel that one of the most important implications of our study is that an inexpensive test such as the PicoGreen method may be used to stratify risk and outcome of patients with cancer more accurately. This is particularly important when one considers the fact that CT imaging is not readily available for lung cancer screening in resource challenged communities and thus the field of cancer screening and diagnosis can potentially significantly benefit from having access to inexpensive liquid biopsy approach such as the technique we are describing in our current manuscript.

Reviewer 2 Report

My take home message from this paper is that one can quantify the ccfDNA using picoGreen rather than going through time-consuming qPCR procedure. While the author showed that the distribution of the samples indeed appear similar using either methods, provided that the primers for mitochondria DNA is used in qPCR, they did not show how well the two methods correlate for individual samples. The author should show a plot of pico-green quantification vs qPCR quantification for a whole group of sample in XY-scatter plot to show there is a strong linear correlation wherever possible, particularly since that appear to be the main point that the authors are arguing for.

The second point is more technical. It's not clear how the qPCR results are converted into pg/100 uL of serum. Please clarify with the equation of conversion. Due to the high number of copies of mitochondria DNA in the cell (hundreds to thousands), it is not impossible purely due to chance that a mitochondria gene is detected at higher frequencies than a nuclear DNA. qPCR data for regular healthy cells would also show similar trend. Therefore it is not absolutely correct to state that most of the ccfDNA in the EV is of mitochondria origin. Rather it should be stated that the mitochondria DNA is more easily detected in qPCR. The vast difference in the sizes of nuclear genome and mitochondrial genome can contribute to this false sense that mitochondria DNA is the majority of total DNA. 

I also have some questions about the statistical methods used. I am really surprised in Figure 4B, there is no difference between CTR and the mTBI group. Closer reading of the method section led me to question the use of Mann-Whitney. Are the sample paired? Are the samples normalized to another gene? If the ccfDNA values (pg/100 uL of serum) is an absolute value calculated from the Ct values, then perhaps student's t-test would suffice. Again, this is why a clear description of how the quantification is done is necessary for the readers to be able to judge if the appropriate statistical method has been applied. 

Minor typos: legends of Figure 3D and 3E. in line 281 and 283.

line 249, delete "were"

line 338, "shown"

line 345, "generation"

Author Response

My take home message from this paper is that one can quantify the ccfDNA using picoGreen rather than going through time-consuming qPCR procedure. While the author showed that the distribution of the samples indeed appear similar using either methods, provided that the primers for mitochondria DNA is used in qPCR, they did not show how well the two methods correlate for individual samples. The author should show a plot of pico-green quantification vs qPCR quantification for a whole group of sample in XY-scatter plot to show there is a strong linear correlation wherever possible, particularly since that appear to be the main point that the authors are arguing for.

Response: We are thankful for this comment. In the revised manuscript we included simple linear regression analysis of the amount of DNA quantity using PicoGreen and by qPCR using mtCOXIII specific primers in a TBI cohort. Despite a relatively small number of analyzed subjects, the calculated R2=0.41 is significant further supporting our conclusion about the direct correlation between DNA calculated with qPCR using mtDNA-specific primers and PicoGreen. In the revised manuscript, the results of our additional analysis as Figure S2.

The second point is more technical. It's not clear how the qPCR results are converted into pg/100 uL of serum. Please clarify with the equation of conversion. Due to the high number of copies of mitochondria DNA in the cell (hundreds to thousands), it is not impossible purely due to chance that a mitochondria gene is detected at higher frequencies than a nuclear DNA. qPCR data for regular healthy cells would also show similar trend. Therefore, it is not absolutely correct to state that most of the ccfDNA in the EV is of mitochondria origin. Rather it should be stated that the mitochondria DNA is more easily detected in qPCR. The vast difference in the sizes of nuclear genome and mitochondrial genome can contribute to this false sense that mitochondria DNA is the majority of total DNA. 

Response: We are thankful to the reviewer for this comment. In the revised manuscript we explained in detail how the calculation was performed. Initially, we used a different known concentration of isolated mtDNA and performed qPCR with mtDNA-specific primers to generate a standard curve where for each concentration (pg) we obtained a specific Ct value. Next, the total DNA was isolated from a cell-free serum followed by qPCR and based on the obtained Ct values we calculated the amount of DNA, and expressed as pg/100ml of serum. The generation of a standard curve for the nuDNA was performed using a similar approach except that we did not isolate nuDNA, instead we used the total DNA since mtDNA contain less than 5% of the total DNA. In fact, both approaches generated similar data because it does not matter whether we used isolated mtDNA followed by qPCR with mtDNA-specific primers or total DNA followed by qPCR with nuDNA primers since certain the amount of DNA results in a particular Ct value. Regarding the comment that most of the DNA in EVs have mitochondria origin. We mentioned in the discussion that although our approach cannot rule out the possibility that due to multiple copies of mtDNA we calculated more mtDNA than nuDNA, but we believe it is rather unlikely. Please note that we used two different primer sets for nuDNA (ACTB and SIRT1) and the amount of calculated nuDNA were several orders of magnitude different between each of the primer sets, even though both genes have two copies. In contrast, two different primer sets specific for mtDNA generated similar data. These observations will suggest release to an extracellular space specific fragments/uneven distribution of nuDNA in opposite to mtDNA where regardless of primers location similar data is obtained. Also as discussed, we cannot rule out that some amount of nuDNA is in the free-floating form and thus is subjected to the DNases present in serum that causes its degradation over time. As we stated in the discussion, additional experiments such as DNAseq are warranted to precisely calculate the amount of mtDNA and nuDNA in circulating EVs, but this is outside the scope of this report. However, along with reviewer’s suggestions, we change the revised manuscript to reflect that mtDNA can be easily detected by qPCR while elaborating on potential reasons.

I also have some questions about the statistical methods used. I am really surprised in Figure 4B, there is no difference between CTR and the mTBI group. Closer reading of the method section led me to question the use of Mann-Whitney. Are the sample paired? Are the samples normalized to another gene? If the ccfDNA values (pg/100 uL of serum) is an absolute value calculated from the Ct values, then perhaps student's t-test would suffice. Again, this is why a clear description of how the quantification is done is necessary for the readers to be able to judge if the appropriate statistical method has been applied. 

Response: We apologize for not making this clear, in the revised manuscript we added detailed explanation on how data was analyzed. Briefly, we first performed normality test of obtained data and based on our analysis in each analyzed data sets at least one of the groups were not normally distributed. Thus, we used Kruskal-Willis test, as a most appropriate test, to determine the significance between groups including data showed in Figure 4B.

Minor typos: legends of Figure 3D and 3E. in line 281 and 283.

line 249, delete "were"

line 338, "shown"

line 345, "generation"

Response: We apologize for any inconvenience. The revised manuscript was carefully edited including complete proofreading for English grammar and spelling.

Round 2

Reviewer 1 Report

The author modified the mansucript and addressed most of the comments of mine. As a clinical application research, the data is meaningful in its field. It is worthy for publication at this stage.